# Proximal Caries Detection Using Short-Wave Infrared Transillumination at Wavelengths of 1050, 1200 and 1300 nm in Permanent Posterior Human Teeth

**DOI:** 10.3390/diagnostics13203257

**Published:** 2023-10-19

**Authors:** Katrin Heck, Karl-Heinz Kunzelmann, Elias Walter, Dalia Kaisarly, Lea Hoffmann, Friederike Litzenburger

**Affiliations:** 1Department of Conservative Dentistry and Periodontology, LMU University Hospital, LMU Munich, 80336 Munich, Germany; karl-heinz@kunzelmann.de (K.-H.K.); elias.walter@med.uni-muenchen.de (E.W.); kaisarly@dent.med.uni-muenchen.de (D.K.); soechtig@dent.med.uni-muenchen.de (F.L.); 2Department of Orthodontics and Dentofacial Orthopedics, LMU University Hospital, LMU Munich, 80336 Munich, Germany; lea.hoffmann@med.uni-muenchen.de

**Keywords:** X-ray microtomography, sensitivity and specificity, reproducibility of results, dental caries, diagnostic imaging, short-wave infrared transillumination

## Abstract

This in vitro study aimed to investigate the diagnostic potential of short-wave infrared transillumination (SWIRT) at 1050, 1200 and 1300 nm for the detection of proximal caries in molars and premolars. It was compared to the diagnostic performance of bitewing radiography (BWR) and micro-computed tomography (µCT) as the reference standard. 250 sound or decayed proximal surfaces of permanent posterior extracted teeth were examined using (1) SWIRT at 1050, 1200 and 1300 nm with two camera systems of different resolutions, (2) BWR and (3) µCT. Thresholds were defined for both test methods and the reference standard for caries in general, enamel caries and dentin caries. All images were assessed by two examiners twice, at an interval of two weeks. SWIRT at wavelengths of 1050, 1200 and 1300 nm achieved sensitivity values more than 2.5 times higher than BWR (enamel caries 3.2–4.4 times; dentin caries 3.25–4.25 times) for the detection of proximal caries. Sensitivity values of SWIRT improved with the higher wavelength. No significant difference was found in diagnostic quality between the two camera systems. SWIRT at 1300 nm imaged proximal enamel caries with the highest accuracy, while the physical optimum for transillumination in dentin was located at a lower wavelength (<1000 nm).

## 1. Introduction

The detection of proximal caries, especially in the early stages, is more important today than ever. On the one hand, prophylactic procedures and the use of fluorides have reduced the severity and extent of caries lesions and shifted its predilection side primarily to the less accessible interdental area and the fissures of occlusal surfaces; on the other hand, caries treatment has changed from aggressive cavity preparation to non-invasive methods, such as fluoridation and resin infiltration, as well as minimally invasive restorative techniques [1]. Non-invasive and radiation-free reliable diagnostic methods are needed to detect these early lesions in the restricted proximal space.

In recent years, notable progress has been made in caries diagnostics, allowing dentists to detect proximal caries with more sensitivity and higher accuracy than is possible with conventional methods, namely visual inspection and digital bitewing radiography (BWR). Among the alternative methods, near-infrared transillumination (NIRT) in particular has come into focus for the detection of proximal caries. In previous studies, near-infrared light (NIR) at 780 nm was directed through the tooth via the root to transilluminate the tooth and detect caries [2,3,4,5,6]. By utilizing NIR light, we can leverage the distinctive optical characteristics of dental tissue and carious lesions.

Healthy enamel appears rather transparent to light at a shorter wavelength, while areas affected by caries exhibit increased light scattering and decreased transparency. In our in vitro studies with 780 nm, NIRT has already achieved a sensitivity of around 80% and specificity of around 95% for the detection of proximal caries. For enamel caries, sensitivity was still 57% and 59%, respectively, with consistently high specificity of 93% and 94% [2,4]. Thus, NIRT already has proven to be a reliable non-invasive and radiation-free alternative method to image dental caries. However, an even higher sensitivity would be desirable to detect caries at an early stage. The application of wavelengths higher than 780 nm promises a solution since the scattering in sound enamel follows the Rayleigh scattering principle and diminishes as the wavelengths increase [7]. At 1300 to 1550 nm, it decreases by a factor of 30 compared to the visible range, reaching its minimum attenuation coefficient at 1300 nm of 3.1 cm^−1^ [8]. In transillumination, this significantly improves the difference between carious and healthy enamel. Although the scattering continues to decrease with increasing wavelengths, the attenuation coefficient increases again above 1310 nm due to the increasing absorption coefficient of water, which is present at 12% in the volume of enamel [8]. Therefore, the investigation of 1000 to 1300 nm for transillumination is highly interesting from a physical point of view.

At present, available devices utilizing NIRT use wavelengths limited to 780 nm [9]. One reason for this is the patenting of wavelengths above 795 nm for transillumination. The other reason is the limitation due to the far more inexpensive camera technology used, which contains silicon sensors and theoretically processes merely wavelengths from 700 to 1000 nm, i.e., in the NIR range [10]. Recent developments in sensor technology, as well as increased demand for infrared cameras for industrial purposes, have reduced the cost of indium gallium arsenide (InGaAs) sensors with the capacity to process wavelengths above 1000 nm. Therefore, the short-wave infrared (SWIR) range starting beyond 1000 nm has become financially more feasible for caries diagnostics. There is a wide range of SWIR cameras available from multiple manufacturers, varying in terms of cost and resolution quality. Currently, cameras with lower resolution cost only one-third as much as those with higher resolution. To date, there is no evidence regarding the best resolution for short-wave infrared transillumination (SWIRT) and its impact on diagnostic performance. Although these higher wavelengths are promising, to date, there is no substantial in vitro study with sufficiently high sample sizes investigating the diagnostic validity of transillumination for proximal caries in human molars and premolars at wavelengths above 1000 nm.

Thus, this study aimed to examine SWIRT at 1050, 1200 and 1300 nm for the detection of early proximal caries using two InGaAs cameras of different resolutions and to determine whether the diagnostic results obtained in each mode differ from those obtained with BWR.

Since BWR is the most commonly used diagnostic tool to aid visual examination in caries detection and has been reported to have a high specificity of 89% in vitro paired with a sensitivity of 42% (in vivo 97% and 24%, respectively) for the detection of proximal caries, it appeared to be the appropriate methodological standard for comparison with SWIRT [11]. Micro-computed tomography (µCT) was used as the reference standard. Based on previous studies of NIRT at a wavelength of 780 nm, detection sensitivity was roughly 60% for enamel caries and 80% for dentin caries, with corresponding high specificities of 93% and 94%, respectively [2,4]. Given that higher wavelengths may result in improved sensitivity, we formulated the following working hypotheses for this experiment (H1): First, SWIRT at 1050, 1200 and 1300 nm has the potential to detect early proximal caries lesions with high sensitivity in enamel (>60%) and dentin (>80%), as well as generally high specificity (>80%). Second, sensitivity values increase with increasing wavelength of light and third, the SWIR camera with the higher resolution is associated with higher accuracy in the detection of proximal caries.

## 2. Materials and Methods

### 2.1. Sample Size

The sensitivity for proximal caries detection in enamel using NIRT is assumed to be approximately 60% [2,4]. We aimed to increase this sensitivity to 80% at wavelengths from 1050 to 1300 nm, with a power of 80%, an alpha of less than 0.05 and an assumed caries prevalence of 50%. Therefore, the sample size calculation resulted in a minimum required sample size of 92 teeth. Since we have used a sample size of 250 samples in previous studies on near-infrared reflectance and BWR, we also increased the sample size to 250 to obtain more robust data with better comparability [12,13,14].

### 2.2. Tooth Selection and Sample Preparation

The experimental procedures conducted on anonymous human dental specimens received waiver approval from the Medical Faculty Ethics Committee at Ludwig Maximilians University in Munich, Germany (KB 20/019).

A total of 250 extracted permanent human molars and premolars were chosen from anonymous patients residing in the Munich area. These teeth exhibited no visible structural changes or damage, except for carious lesions, and had not undergone any restorative procedures. Additionally, the teeth had fully matured roots and were visually examined to ensure their suitability. The International Caries Detection and Assessment System 2 (ICDAS II) was employed to evaluate one proximal surface of each sample in direct view, resulting in a balanced composition of healthy (ICDAS 0 = 125) and carious (*n* = 125) samples [15]. The carious specimens consisted of 18 with an ICDAS score of 1, 69 with a score of 2, 25 with a score of 3, 4 with a score of 4 and 9 with a score of 5.

The teeth were manually cleaned using scalers and assigned identification numbers. Subsequently, the teeth were affixed with composite (Luxatemp Star, DMG, Hamburg, Germany) in 3D-printed specimen holders designed to fit into the tomograph [13]. Throughout the entire process, the teeth were stored in Ringer’s solution containing 2% sodium azide at a temperature of 4 °C, starting shortly after extraction. They were only removed from the storage boxes for a brief period of approximately five minutes during the measurements.

### 2.3. Short-Wave Infrared Transillumination Images

An in vitro model was developed to generate SWIRT images (Figure 1). For this purpose, we constructed a 3D-printed holder for the samples into which a split fiber optic cable (BFY1000LS02, Thorlabs, Newton, NJ, USA) was inserted from two sides to illuminate the samples. Three different light-emitting diodes (LEDs) were used as light sources for the wavelengths of 1050, 1200 and 1300 nm (M1050L4, M1200L3 and M1300F1, Thorlabs, Newton, NJ, USA) to be investigated. SWIRT images of each specimen were acquired with two SWIR cameras in occlusal view using a 50 mm lens (LM50HC-SW, KOWA, Nagoya, Japan). One camera had a lower resolution (WiDy Sens 320V-ST, resolution 320 × 256, New Imaging Technologies, Verrières-le-Buisson, France) and the other one a higher resolution (Wildcat-640-20-100-USB, resolution 640 × 512, XENICS, Leuven, Belgium). All specimens were photographed in a darkened room, and the equipment was also housed in a black box to reduce stray light from the environment.

### 2.4. Digital Bitewing Radiography

To ensure that the pixel grayscale distribution corresponded to clinical bitewing radiographs, a well-established phantom model was used for the in vitro radiography [13]. To avoid artifacts due to overlap, specimens were positioned during radiography without contact with adjacent teeth. A Heliodent DS dental X-ray unit (Sirona, Bensheim, Germany) was used for the exposures with specific settings: 60 kV, 7 mA, 200 mm FHA cone and an exposure time of 0.08 s. In addition, a digital charge-coupled device (Intra-Oral II CCD sensor, Sirona, Bensheim, Germany) measuring 30.93 × 40.96 × 7.0 mm was used to capture the images.

### 2.5. Micro-Computed Tomography

All specimens were scanned using a fully shielded cone-beam desktop microcomputed tomograph (Sanco Medical, Bassersdorf, Switzerland) with the parameters of 70 kV and 114 µA. The field of view of the tomograph was limited to 16.5 mm. During scanning, each sample was positioned in the center of a water-filled cuvette. The raw scan (RSQ) data sets included voxels with a side length of 32 µm and a scan resolution of 512 × 512 points subsequently reconstructed into 3D data sets (ISQ files). To facilitate further image processing, the plugin “KHKs_Scanco_ISQ_FileReader” was used to import the data into Fiji [16,17].

### 2.6. Calibration and Training for the Evaluation of Findings of All Index Test Methods and the Reference Test Method

The examiners (KH and FL) were trained and calibrated under the supervision of a trainer (KHK) with 37 years of diagnostic experience in research and daily clinical business at University Hospitals in Germany.

The training included theoretical instruction on the principles and the scoring system of all three methods, as well as a discussion and explanation of exemplary image sets for each score and method. Subsequently, the assessment with the explained scoring systems was practiced based on new image sets. In the case of different diagnostic decisions, the potential errors and artifacts for each method were identified and discussed, and strategies were worked out together.

The examiner’s reliability assessment of the final evaluation session resulted in an agreement greater than 90% for the intra- and inter-reliability assessment.

### 2.7. Evaluation of Findings of the Index Test Methods and the Reference Test Method

The evaluation of all of the images (SWIRT, BWR and µCT) was performed by the two calibrated investigators (KH and FL) independently under standard conditions in two cycles at least two weeks apart. For the ratings that did not match, a consensus was reached. All assessments were performed in a darkened room, north-facing window (blinds closed 2/3) on a calibrated monitor with a sitting distance of 60 cm (arm’s length) and eyes adjusted to lighting conditions for at least 5 min [12,13,14].

The scoring of the SWIRT images followed the classification published by Lederer et al., where code 0 represents the absence of a lesion, code 1 indicates a lesion visible in the enamel, code 2 is a lesion in the enamel with a point contact to the dentin–enamel junction (DEJ), code 3 is a lesion with linear contact to the DEJ and code 4 is a visible lesion extending into the dentin. [4]. For statistical analysis, these scores were summarized according to the diagnostic thresholds’ enamel caries (code 1 or 2) and dentin caries (codes 3 or 4) (Table 1). All surfaces that were not scored with code 0 were classified as carious surfaces (codes 1, 2, 3 or 4). Additionally, it was checked whether a clear distinction between enamel and dentin according to the DEJ was possible when assessing the proximal region (0—no, 1—yes).

For BWR findings, the classification according to Marthaler et al. was applied, where 0 represents the absence of radiolucency, 1 and 2 represent the presence of radiolucency in the outer and inner enamel halves and 3 and 4 represent its presence in the outer and inner dentin halves [18].

The µCT data sets were evaluated using segmentation and automatic centerline determinations for enamel and dentin [13]. The proximal surfaces were thus evaluated in a semiquantitative system, analogous to the evaluation of BWR; the absence of radiolucency was scored 0, the presence of radiolucency in the outer and inner half of the enamel was 1 and 2 and in the outer and inner half of dentin was scored 3 and 4.

### 2.8. Statistics

For sample size calculation, SAS/STAT software (SAS/STAT, Version 15.1, Cary, NC, USA), the Proc Power procedure, was used.

Table 1 shows the diagnostic thresholds for the test and reference methods for the classification sound surface, enamel caries and dentin caries.

Statistical analysis was performed using the software SPSS (IBM SPSS Statistics for Windows, Version 27.0, Armonk, NY, USA) and Excel (Excel Version 16.78, Microsoft, Redmond, WA, USA) and included the calculation of sensitivity, specificity and area under the curve (AUC). Multiple comparisons of the AUCs within the thresholds were performed using easyROC [19]. The interpretation of AUC values was according to Hosmer and Lemeshow [20]. The receiver operating characteristic (ROC) curve and the corresponding AUC value were used to estimate the overall diagnostic performance of all imaging techniques.

The proportion of correctly classified surfaces out of the total of all 250 assessments was referred to as the overall accuracy (ACC) for each method.

Cochran’s Q test was used to compare the visibility of the DEJ at different wavelengths and camera types.

Reliability assessments for the categorial variables SWIRT, BWR and µCT were calculated using linearly weighted Cohen’s kappa (wk), where a 1-category difference was considered less severe than a 2-category difference. All kappa values were interpreted according to Landis and Koch [21].

The two-sided significance level was set at α = 0.05 for all tests. Bonferroni correction was used for multiple testing.

## 3. Results

The reference method µCT achieved almost perfect agreement in inter-examiner (wk 0.99, CI: 0.98–1.00) and intra-examiner (wk 0.99, CI: 0.98–1.00; wk 0.96, CI: 0.93–0.98) scores [21]. The reliability assessment showed almost perfect agreement for SWIRT and BWR (Table 2).

The results of the occlusal examinations at different wavelengths and with two different camera types, as well as the BWR assessment compared to µCT as a reference test, are shown in Table 3.

As wavelengths increased, the ability of SWIRT to detect the DEJ improved significantly (resolution 320 × 256 at 1050 nm, 75.6%; at 1200 nm, 82.4%; at 1300 nm, 90.4%—resolution 640 × 510 at 1050 nm, 73.2%; at 1200 nm, 89.2%; at 1300 nm, 93.6%) (*p* < 0.05).

The overall accuracy of SWIRT ranged from 86 to 93%, with higher accuracy observed as wavelengths increased within each lesion type. In comparison, BWR achieved statistically significantly lower accuracy (between 74% and 82%) compared to SWIRT (*p* < 0.05). Sensitivity values for SWIRT to detect caries in general and dentin lesions reached high values of over 80% at a wavelength of 1300 nm, associated with high specificity values of over 90%. For enamel lesions, the sensitivity values were between 40 and 53%, also paired with high specificity (Table 4). The conventional bitewing imaging comparison group achieved notably lower sensitivity across all lesion types.

ROC curves are displayed in Figure 2. All AUC values for SWIRT ranged from 0.80 to 0.91 for caries lesions as well as dentin caries and are thus considered excellent discrimination. In contrast, for enamel caries, AUC values reached around 0.70, which corresponds to acceptable discrimination. For BWR, the values for all caries categories were only between 0.54 and 0.65 and are therefore considered only poor to good discrimination (Table 4) [20].

All transillumination testing modalities showed statistically significantly higher values for dentin and caries thresholds using comparisons of AUC with BWR (*p* < 0.001), though they did not differ statistically significantly among each other (*p* > 0.05). For enamel caries, a significant difference in the AUC of all transillumination modalities was observed for the resolution of 640 × 510 at 1300 and 1050 nm to BWR. There were no significant differences in enamel caries between the different wavelengths and camera types. With increasing wavelength, enamel becomes more translucent and the dentin darker (Figure 3, Figure 4 and Figure 5). Enamel lesions are most conspicuous at 1300 nm. Advanced carious lesions are more frequently detected in the dentin tissue at 1050 nm than at 1300 nm (Figure 5).

## 4. Discussion

This study aimed to assess the potential of SWIRT at 1050, 1200 and 1300 nm for early proximal caries detection. Two InGaAs cameras of varying resolutions were employed, and their diagnostic results were compared to those acquired for BWR serving as the test method and µCT examination serving as the reference standard.

Previous in vitro studies and clinical trials have demonstrated the promising diagnostic accuracy of transillumination at 780 nm when assessing teeth from the occlusal direction [2,4,22]. As reported in other studies, increasing wavelengths above 780 nm should significantly improve the diagnostic potential of transillumination [23,24]. The understanding of light propagation in human dental hard tissues at higher wavelengths has primarily been gained from tests on tooth sections, instead of on whole teeth. Whole tooth samples exhibit altered light propagation characteristics compared to thin slices due to aspects such as mass, the convex shape of surfaces, anatomical variations or structural irregularities.

Preliminary clinical investigations show that using the 1300 nm wavelength in transillumination enhances the diagnostic efficacy of identifying proximal caries more than using the prior 780 nm wavelength [23,24,25]. This improvement surpasses that which is achieved through BWR, reaching sensitivity and specificity values of 63% and 62%, respectively [24]. It should be noted, however, that these studies are limited so far to a single group of researchers. Their investigations are restricted to premolars scheduled for extraction due to orthodontic needs or rely on BWR as the reference standard [23,24,25]. Although clinical diagnostic studies are essential for a solid scientific statement about novel methods and possess higher relevance than in vitro studies, they exhibit limitations, lacking a valid reference standard free of compromises. Common approaches to validate carious lesions in vivo, such as assessment of surfaces after tooth separation, ex vivo examinations or examinations using BWR as the reference standard, compromise the underlying sample pool or show incomplete validity. In vitro analyses, on the other hand, are carried out in idealized laboratory setups and therefore have much weaker significance than in vivo studies. It is important to conduct in vitro studies, as a precisely selected sample pool with even distribution between molars and premolars, as well as an assessment of carious lesions within enamel and dentin, can provide strong statistical conclusions about the method’s sensitivity and specificity. In vitro studies on caries detection testing new methods with a comparatively large number of samples validated by histology or μCT are rare. This is an advantage of the present study because μCT is an expensive procedure but provides a solid reference standard without destroying the samples. The high sample size and subsequent statistical analysis of the accuracy of SWIRT at 1050–1300 nm for the detection of proximal caries in whole human posterior teeth summarizes the innovative nature of this study. There are diagnostic studies using SWIR cameras with different resolutions, but none of these have directly compared camera types with different resolutions [23,24,25,26]. Therefore, in this study, the SWIR images of each surface were taken by two InGaAs cameras with different resolutions. This direct comparison gives information about the influence of the camera type on the diagnostic performance of SWIRT. This study’s quality is supported by impressive SWIR images that highlight the potential of this method, emphasizing the need for further research in this field.

The outcomes of this study are consistent with previous research findings and ascribe SWIRT the highest potential to identify caries lesions, with a maximum of 84% for caries in general, 53% for enamel and 85% for dentin, while specificity was around 95% for all diagnostic categories. These findings are similar to or slightly lower than the results reported at 780 nm by Lederer et al. and necessitate a partial revision of the initial hypothesis; transillumination at 1050, 1200 and 1300 nm does not possess the potential to detect early proximal caries lesions with sensitivity values over 60% for enamel and 80% for dentin [2,4]. The reasons for the unexpectedly lower sensitivity values of SWIRT, i.e., the underestimation of proximal tooth caries, comprise several factors, e.g., sample size and composition, the design of the in vitro model, the different tooth anatomies of the samples and the characteristics of light propagation in the tooth hard tissue. Having a larger sample size of 250 extracted teeth in this study increases the diagnostic power, significance and validity of the findings.

One important aspect that may have determined diagnostic accuracy towards underestimation, for SWIRT as well as for BWR, is the unknown level of demineralization of the 18 lesions in the outer half of the enamel. It is known that the scattering of initial non-cavitated enamel lesions intensifies with increasing levels of demineralization [9]. In our analysis, the demineralization may not exceed the critical point of mineral loss required to visualize a lesion using transillumination but is still large enough to be identified as diseased by the µCT. Further studies with a focus on comparing the propagation of light in de- and remineralized tooth structures are important for learning more about this and are currently not sufficiently available.

Moreover, diagnostic decisions in SWIRT images are influenced by aspects concerning the in vitro model. The sample positioning between the light source and the camera has an influence on the illumination of the specimen, as well as on the imageability of some initial caries lesions. The appearance of an enamel lesion alters depending on the angle at which the light is directed onto the demineralized area. For example, the contact of a lesion to the DEJ may be present on first viewing but then may disappear after the light source is rotated. The preparation of the images is therefore always preceded by a dynamic examination of the tooth, during which the optimum angle of light incidence is determined after inclining and rotating the specimen. In addition, it can be observed that live images often show proximal surfaces with higher precision and better exposure than single images. For this analysis, only single-shot images were used without exception. Another aspect was an even distribution of light within the crown of the tooth. Premolars, for example, were often overexposed, while in the case of more massive molars, light could only be introduced selectively, and parts of the crown remained dark (Figure 3b,c). It was not possible to obtain the optimal image for every sample and wavelength. In clinical use, the tooth is embedded by gum and alveolar bone, which might be responsible for a more balanced circular light transmission into the tooth. In preliminary experiments, we attempted to simulate these anatomical tissues using a liquid containing dissolved hydroxyapatite crystals and silicate beads of 0.5 mm diameter, in which the root side of the tooth was mounted. Since there was no noticeable difference in illumination quality with SWIR light, it remains to be speculated whether real bone tissues would affect the image quality and the diagnostic accuracy of SWIRT. This should be the subject of further studies.

Anatomical aspects of the samples influence the diagnostic accuracy of transillumination as well. Since the donors of the samples are anonymized for ethical reasons, nothing is known about age of the dental material. It has been found that the crystalline structure of enamel expands with age, which might be associated with a general increase in scattering in the affected tissues [27,28]. Sensitivity values for dental decay are also significantly influenced by the localization of caries lesions in the apical direction, as the light is attenuated on its passage through the tissue [29]. Since the average path length of light through healthy enamel at 1310 nm is less than 4 mm, lesions that are cervically closer to the DEJ cannot be directly detected by transillumination (Figure 4c,f,i) [30]. The degree of scattering of enamel varies from individual to individual and is particularly high in teeth with mineralization irregularities. Teeth with hypo- or hypermineralization exhibit a strong increase in scattering, whereby the tissues generally appear brighter. Thus, the optical discrimination between sound and demineralized areas becomes more difficult. Similar observations can be made when changes in occlusal enamel morphology are present. Accessory cusps or enamel defects in the marginal ridge area increase opacity and can cause false positive results (Figure 3).

From experiments of transillumination on tooth sections, we learn that an increasing wavelength leads to a lower attenuation coefficient and thereby less scattering and higher transparency in sound enamel [9,31,32]. The scattering of demineralized enamel is thus associated with a higher lesion contrast. This phenomenon is validated by the results of this study. Therefore, the second hypothesis meaning that sensitivity of SWIRT for the detection of proximal caries in general, in enamel and dentin increased with increasing wavelength is confirmed. However, these results must be seen in the context of the scoring system used to classify the images into five different categories 0 to 4. Four of the five categories (scores 0 to 3) define changes in enamel, while only one refers directly to changes in dentin (score 4). Therefore, the calculation of diagnostic accuracy is only indirectly based on information about the potential of SWIRT to show caries in dentin at these higher wavelengths. Regarding the images, we observe that nearly all surfaces assessed as having direct dentin involvement (score 4) are dentin caries according to the reference standard. Further, in almost all of the samples, dentin becomes darker with increasing wavelength (Figure 3, Figure 4 and Figure 5). This is also reflected by statistical data about the discrimination between sound enamel and dentin along the DEJ. With increasing wavelength, the discrimination between enamel and dentin improves for about 20%. At 1300 nm, the dentin nucleus was clearly identifiable from the enamel in over 90% of all specimens. Regarding the images (Figure 3, Figure 4 and Figure 5), this improvement is not only due to the increasing transparency of the enamel but also to the higher opacity of dentin. Figure 5 compares four images of one dentin caries lesion using wavelengths from 780 to 1300 nm. While at 780 nm, a caries lesion can be discriminated from sound dentin, this is no longer possible at higher wavelengths. The increasing attenuation coefficient of dentin is dominated by the absorption of water, which was stated to be highest at 1450 nm [33]. This phenomenon also becomes obvious when regarding the image of Figure 3c. Within the scope of our investigation, we initially intended to generate SWIR images at 1550 nm, but the absorption of water in dentin at this wavelength led to strong degradation of image quality (Figure 3c). Therefore, it can be stated that for lower attenuation of light in dentin, its physical optimum for transillumination is more in the range of lower wavelengths (<1000 nm). The images in Figure 4h and Figure 5a are images from our archive that were not part of the statistical analysis.

In this project, SWIR images are generated with wavelengths from 1050 nm to 1300 nm. For wavelengths above 1000 nm, cameras with InGaAs sensors are needed, as cameras with silicon sensors only have sufficient quantum efficiency for wavelengths up to 800 nm. The InGaAs cameras currently on the market are significantly more expensive than cameras with silicon sensors and were available with two possible resolutions (320 × 256 pixels and 640 × 512 pixels) at the time this study was planned, whereby the better resolution was associated with a significant price increase. In this experimental setup, we compared both camera types to find out how the resolution affects the diagnostic performance and thereby the cost structures for possible further development in clinical use. It was hypothesized that cameras with better resolution and significantly higher price levels would contribute to higher diagnostic accuracy. The analysis showed no significant difference between the two InGaAs cameras regarding their diagnostic performance. Although the camera system with the higher resolution provides images with higher contrast, it requires a significantly higher financial investment. At this point, it can be concluded that the use of a camera system with a resolution of 320 × 256 pixels is sufficient for proximal caries detection with transillumination above 1000 nm. It is crucial to reassess and re-evaluate these findings on a regular basis as new camera types and technical innovations are constantly being developed.

The results for the diagnostic performance of the test method, BWR, are significantly less sensitive than for SWIR imaging at all three wavelengths. The accuracy of BWR, especially the values of sensitivity, is lower than that reported in a review by Schwendicke et al. [11]. This is mainly caused by our sample pool considering 18 early enamel lesions as well as 47 early dentin lesions (Table 3). Although these lesions are detectable by µCT scans, the mineral loss does not reach the extent to be visualized by digital radiology [34,35].

Since the sample pool predominantly comprises surfaces that are sound or with non-cavitated lesions, it can be concluded that the detection of early caries lesions is possible with the advent of transillumination, in contrast to BWR. The use of SWIRT offers a broader range of information on the early stages of proximal caries that remain undetected using BWR. As for interpreting this insight in the clinical context, this information gain must be interpreted correctly to avoid overtreatment of carious lesions. The higher sensitivity of SWIR imaging can support dental practitioners to detect early lesions and to carry out appropriate prophylactic measures. Of course, light optical diagnostics cannot replace the visual examination or claim to be able to assess the general caries risk of a patient. One major shortcoming of transillumination imaging is that it cannot differentiate cavitated from non-cavitated proximal lesions. Cavitations are an important indicator for the progression of a lesion into dentin [36,37]. Abdelaziz et al. showed that cavitations on proximal surfaces can be detected with NIRT at 780 nm using a dye solution [38]. In a similar experiment, the detection of cavities without imaging software might be possible at wavelengths over 1000 nm.

In previous studies on transillumination of teeth with 780 nm, the light scattering properties of the tooth surrounding tissues were mimicked using a colloid to reduce the laser speckle effect, which is responsible for granular and blurred image quality [2]. Since we used LEDs in this in vitro setup, the speckle effect played no role here. We achieved an optimal transillumination of the samples using a 3D-printed sample holder with an inserted split fiber optic cable. With this strategy, it was possible to create SWIR images of whole extracted teeth which have not yet been shown in the literature with comparable quality and clarity.

## 5. Conclusions

In this study, we successfully developed an in vitro model for short-wave infrared transillumination to image permanent posterior teeth from the occlusal direction at wavelengths of 1050, 1200 and 1300 nm, with high quality and almost no artifacts (Figure 4 and Figure 5). Compared to BWR, SWIRT reveals an almost three times higher sensitivity for proximal caries in general, enamel and dentin. Additionally, SWIRT exhibits consistent high specificity.

With increasing wavelength, sound enamel becomes more transparent, while caries lesions are visible with greater precision, providing improved sensitivity in the detection of interproximal caries (Figure 4 and Figure 5). Due to scattering and absorption properties in dentin, it appears darker with increasing wavelengths, while lesions in dentin become less visible (Figure 5). For the detection of proximal caries in enamel, the light has its physical optimum at 1300 nm, while in dentin, it seems to remain below 1000 nm.

Furthermore, our study suggests that employing a more cost-effective camera system with a resolution of 320 × 256 pixels is currently sufficient for reliable clinical research diagnosis of proximal caries detection using transillumination at wavelengths exceeding 1000 nm.

## Figures and Tables

**Figure 1 diagnostics-13-03257-f001:**
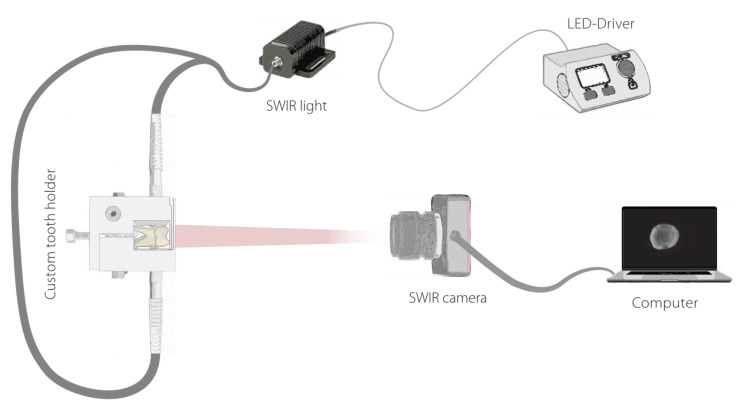
Schematic illustration of the in vitro model.

**Figure 2 diagnostics-13-03257-f002:**
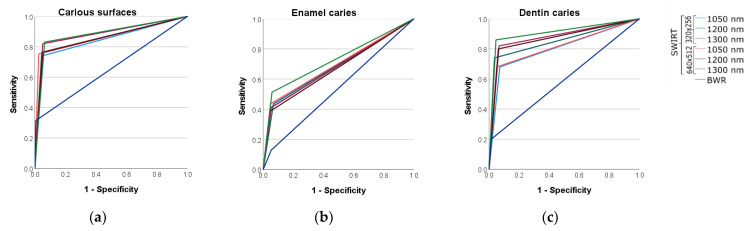
ROC curves for caries (**a**), enamel (**b**) and dentin (**c**) thresholds for all imaging techniques examined.

**Figure 3 diagnostics-13-03257-f003:**
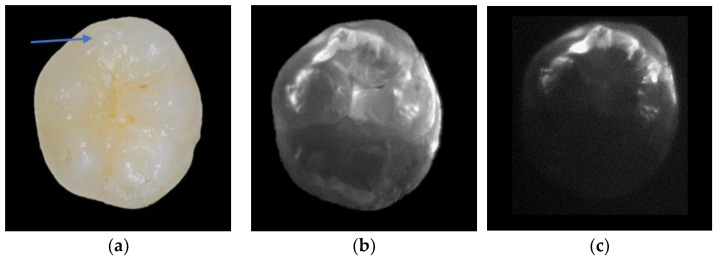
White light image with an accessory cusp in the proximal area marked by an arrow (**a**), which can be misdiagnosed as caries in the SWIRT image at 1300 nm (**b**). This is also visible at 1550 nm, although light attenuation in the dentin is dominant at this wavelength (**c**).

**Figure 4 diagnostics-13-03257-f004:**
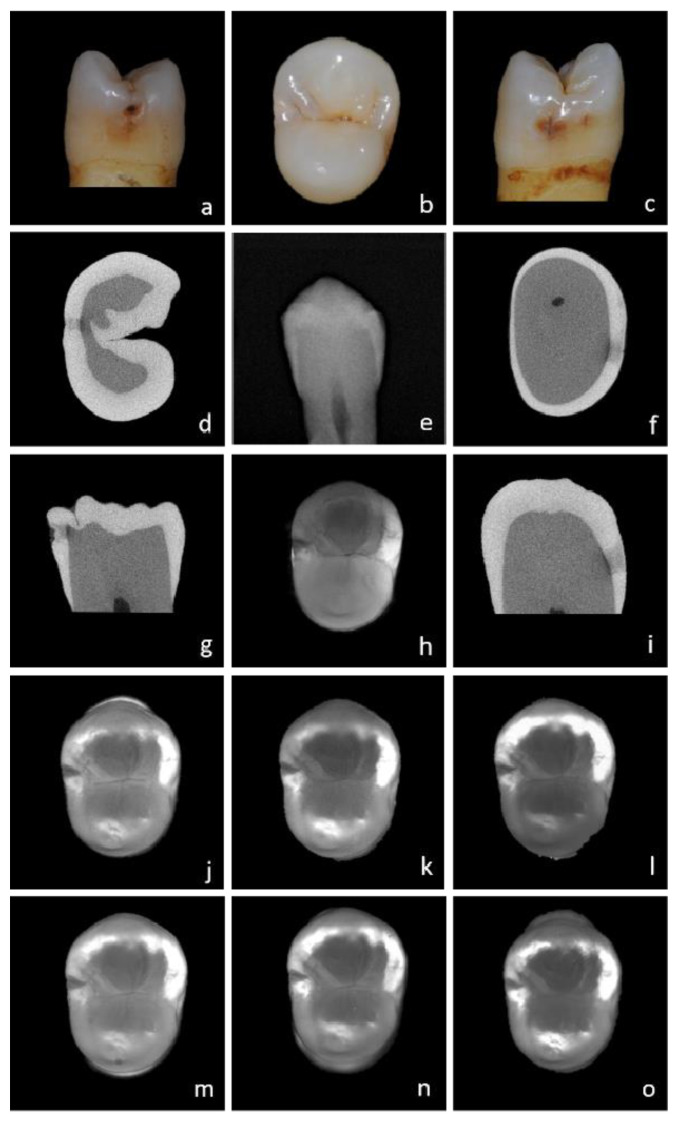
White light images of a sample in mesial (**a**), occlusal (**b**) and distal (**c**) view. Transversal section in µCT of the mesial (**d**) and distal (**f**) surfaces, and sagittal section mesial (**g**) and distal (**i**). BWR (**e**). Transillumination images at 780 nm (**h**) and 1050, 1200 and 1300 nm (**j**–**l**) with a lower resolution camera as well as higher resolution camera (**m**–**o**).

**Figure 5 diagnostics-13-03257-f005:**
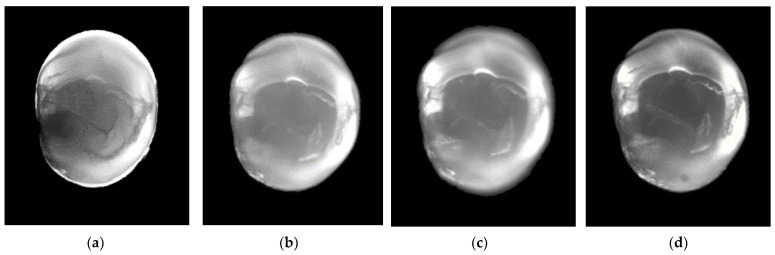
Transillumination images at 780 (**a**), 1050 (**b**), 1200 (**c**) and 1300 nm (**d**). The better discrimination between enamel and dentin can be observed at higher wavelengths. The dentin caries can still be seen in the dentin at 780 nm, while it can no longer be distinguished from the healthy dentin at increasing wavelengths.

**Table 1 diagnostics-13-03257-t001:** Diagnostic thresholds for all imaging modalities examined.

	Sound Surface	Enamel Caries	Dentin Caries
SWIRT	0	1 or 2	3 or 4
BWR	0	1 or 2	3 or 4
µCT	0	1 or 2	3 or 4

SWIRT = short-wave infrared transillumination, BWR = bitewing radiography, µCT = microcomputed tomography.

**Table 2 diagnostics-13-03257-t002:** Inter- and intra-examiner reliability using linear weighted κ-values for short-wave infrared transillumination (SWIRT) and digital bitewing radiography (BWR) evaluation with corresponding 0.95 confidence intervals in parentheses.

			Inter-	Intra-	
	WL	RES	Examiner 1 vs. Examiner 2	Examiner 1	Examiner 2
SWIRT	1050	320 × 256	0.96	0.91	0.92
			(0.93–0.99)	(0.86–0.95)	(0.88–0.97)
		640 × 510	0.95	0.97	0.95
			(0.91–0.99)	(0.94–1.00)	(0.92–0.99)
	1200	320 × 256	0.93	0.96	0.94
			(0.88–0.97)	(0.92–0.99)	(0.90–0.98)
		640 × 510	0.90	0.97	0.94
			(0.84–0.96)	(0.93–1.00)	(0.89–0.98)
	1300	320 × 256	0.94	0.97	0.95
			(0.90–0.98)	(0.95–1.00)	(0.92–0.99)
		640 × 510	0.93	0.96	0.94
			(0.89–0.98)	(0.93–0.99)	(0.91–0.98)
BWR			0.86	0.97	0.85
			(0.78–0.94)	(0.94–1.00)	(0.76–0.93)

WL = wavelength in nm, RES = resolution (pixel).

**Table 3 diagnostics-13-03257-t003:** Cross-table of ratings for short-wave infrared transillumination (SWIRT) for different wavelengths (WL) and resolutions (RES), as well as for digital bitewing radiography (BWR) with corresponding ratings of microcomputed tomography (µCT). The images that were not assessable are marked with na (not applicable).

				µCT					
	WL	RES	Score	0	1	2	3	4	Total
SWIRT	1050	320 × 256	0	148	10	4	11	0	173
			1	2	4	7	3	0	16
			2	3	0	5	4	0	12
			3	4	4	5	22	0	35
			4	0	0	1	7	5	13
			na	0	0	1	0	0	1
		640 × 510	0	153	11	4	9	0	177
			1	0	6	9	2	0	17
			2	2	0	2	7	0	11
			3	2	1	7	24	0	34
			4	0	0	0	5	5	10
			na	0	0	1	0	0	1
	1200	320 × 256	0	150	14	3	4	0	171
			1	1	3	6	3	0	13
			2	4	0	8	7	0	19
			3	2	1	4	30	3	40
			4	0	0	0	2	2	4
			na	0	0	2	1	0	3
		640 × 510	0	149	15	3	4	0	171
			1	1	2	8	1	0	12
			2	2	0	5	6	0	13
			3	5	1	6	35	3	50
			4	0	0	0	1	2	3
			na	0	0	1	0	0	1
	1300	320 × 256	0	149	12	2	2	0	165
			1	2	3	8	2	0	15
			2	2	0	6	5	0	13
			3	4	3	5	36	2	50
			4	0	0	1	2	3	6
			na	0	0	1	0	0	1
		640 × 510	0	147	12	1	2	0	162
			1	5	5	7	0	0	17
			2	2	0	9	6	0	17
			3	3	1	4	35	2	45
			4	0	0	1	4	3	8
			na	0	0	1	0	0	1
BWR			0	155	17	16	31	0	219
			1	0	0	4	3	1	8
			2	1	0	1	6	0	8
			3	0	1	2	6	1	10
			4	0	0	0	0	3	3
			na	1	0	0	1	0	2
			Total	157	18	23	47	5	250

**Table 4 diagnostics-13-03257-t004:** Diagnostic performance of short-wave infrared transillumination (SWIRT) and bitewing radiography (BWR) with corresponding 0.95 confidence intervals in parentheses.

		WL	RES	SE	SP	FP	FN	AUC
SWIRT	Carious lesions	1050	320 × 256	0.73	0.94	0.06	0.27	0.85
				(0.69–0.76)	(0.85–1.03)	(−0.03–0.15)	(0.24–0.37)	(0.79–0.90)
			640 × 510	0.74	0.97	0.03	0.26	0.86
				(0.71–0.76)	(0.88–1.06)	(−0.06–0.12)	(0.24–0.36)	(0.81–0.92)
		1200	320 × 256	0.77	0.96	0.04	0.23	0.87
				(0.73–0.80)	(0.87–1.04)	(−0.04–0.13)	(0.20–0.33)	(0.81–0.92)
			640 × 510	0.76	0.95	0.05	0.24	0.86
				(0.73–0.80)	(0.86–1.04)	(−0.04–0.14)	(0.20–0.33)	(0.80–0.91)
		1300	320 × 256	0.83	0.95	0.05	0.17	0.89
				(0.79–0.86)	(0.87–1.03)	(−0.03–0.13)	(0.14–0.25)	(0.84–0.94)
			640 × 510	0.84	0.94	0.06	0.16	0.89
				(0.80–0.88)	(0.86–1.01)	(−0.01–0.14)	(0.12–0.24)	(0.84–0.94)
	Enamel lesions	1050	320 × 256	0.40	0.94	0.06	0.60	0.68
				(0.37–0.43)	(0.79–1.09)	(−0.09–0.21)	(0.57–0.78)	(0.58–0.79)
			640 × 510	0.43	0.95	0.05	0.58	0.69
				(0.39–0.46)	(0.79–1.10)	(−0.10–0.21)	(0.54–0.76)	(0.59–0.80)
		1200	320 × 256	0.44	0.93	0.07	0.56	0.70
				(0.40–0.47)	(0.77–1.08)	(−0.08–0.23)	(0.53–0.74)	(0.59–0.80)
			640 × 510	0.38	0.95	0.05	0.63	0.67
				(0.35–0.40)	(0.80–1.10)	(−0.10–0.20)	(0.60–0.81)	(0.56–0.78)
		1300	320 × 256	0.43	0.95	0.05	0.58	0.69
				(0.39–0.46)	(0.79–1.10)	(−0.10–0.21)	(0.54–0.76)	(0.59–0.80)
			640 × 510	0.53	0.94	0.06	0.48	0.74
				(0.49–0.56)	(0.78–1.09)	(−0.09–0.22)	(0.44–0.64)	(0.64–0.84)
	Dentin lesions	1050	320 × 256	0.65	0.93	0.07	0.35	0.80
				(0.62–0.69)	(0.80–1.06)	(−0.06–0.20)	(0.31–0.48)	(0.72–0.89)
			640 × 510	0.65	0.95	0.05	0.35	0.81
				(0.62–0.68)	(0.82–1.08)	(−0.08–0.18)	(0.32–0.49)	(0.73–0.90)
		1200	320 × 256	0.73	0.96	0.04	0.27	0.85
				(0.70–0.75)	(0.84–1.09)	(−0.09–0.16)	(0.25–0.41)	(0.78–0.93)
			640 × 510	0.79	0.94	0.06	0.21	0.87
				(0.76–0.82)	(0.83–1.05)	(−0.05–0.17)	(0.18–0.32)	(0.80–0.94)
		1300	320 × 256	0.83	0.93	0.07	0.17	0.88
				(0.79–0.86)	(0.83–1.04)	(−0.04–0.17)	(0.14–0.27)	(0.81–0.94)
			640 × 510	0.85	0.95	0.05	0.15	0.91
				(0.82–0.88)	(0.86–1.05)	(−0.05–0.14)	(0.12–0.25)	(0.85–0.97)
BWR	Carious lesions			0.30	0.99	0.01	0.70	0.65
				(0.29–0.32)	(0.90–1.09)	(−0.09–0.10)	(0.68–0.86)	(0.58–0.73)
	Enamel lesions			0.12	0.95	0.05	0.88	0.54
				(0.09–0.15)	(0.85–1.05)	(−0.05–0.15)	(0.85–1.04)	(0.44–0.64)
	Dentin lesions			0.20	0.98	0.02	0.80	0.59
				(0.18–0.21)	(0.88–1.09)	(−0.09–0.12)	(0.79–1.02)	(0.50–0.69)

WL = wavelength in nm, RES = resolution (pixel), SE = Sensitivity, SP = specificity, FP = false positive, FN = false negative, AUC = area under the curve.

## Data Availability

All data and materials are transparent within the manuscript.

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
