# Peer review of "Proximal Caries Detection Using Short-Wave Infrared Transillumination at Wavelengths of 1050, 1200 and 1300 nm in Permanent Posterior Human Teeth"

_diagnostics, 2023, doi:10.3390/diagnostics13203257_

Round 1

Reviewer 1 Report

In this manuscript, the authors assessed the effectiveness of near-infrared transillumination in detecting proximal caries in molars and premolars, comparing it to bitewing radiography and micro-computed tomography at the wavelengths of 1050, 1200, and 1300 nm. The results presented are commendable, with some issues that need addressing.

Major issues:

1.     Please summarize the innovations of this work, as compared to the previous studies, especially those in the NIRT dental imaging field.

Minor issues:

1.     Page 3, line 98, the author mentioned high specificity without listing affiliation.

2.     Page 3, line 137, double-check the split fiber optic cable with catalog: BYF1000LS02 or BFY1000LS02.

3.     Page 10, lines 309 to 313, the citations are needed.

N/A

Reviewer 2 Report

Dear Authors, 

you made a great work! However, some improvements are suggested before acceptance. 

Reviewer 3 Report

Thank you for inviting me to review the manuscript titled 'Proximal caries detection using near-infrared transillumination at wavelengths of 1050, 1200 and 1300 nm in permanent posterior human teeth'. The manuscript presents an in vitro study assessing the diagnostic capability of caries using different transillumination methods with high frequencies. The study is well-executed, the sample is certainly representative, the methods are well-described, and the results are interesting and significant. To enhance the study, the authors should address the following points:

1.     The authors state that obtaining ethical committee approval was not necessary, but then they mention an acronym (KB 20/019) that seems to refer to an ethical committee act (lines 113-115). Could they provide further clarification?

2.     The diagnostic capabilities of bitewing radiography in this study were found to be particularly low. For example, DOI: 10.1016/j.jdent.2015.02.009 reported higher values. How do the authors explain this?

3.     Figure 2 is in German. Please translate it into English.

4.     Were there any statistical comparisons of sensitivity/specificity between NIRT and BWR based on different lesion depths while keeping tissue constant?

Round 2

Reviewer 3 Report

The authors have satisfactorily addressed the requests. I recommend accepting the manuscript

Author Response

Thank you for the positive vote and the work you put into the review.
